# Use of unmanned aerial vehicles (UAVs) for mark-resight nesting population estimation of adult female green sea turtles at Raine Island

Andrew Dunstan[1]*, Katharine Robertson[1], Richard Fitzpatrick[2], Jeffrey Pickford[3], Justin Meager[1]

**1** Queensland Department of Environment and Science, Brisbane, Queensland, Australia, **2** Biopixel Oceans Foundation, James Cook University, Smithfield, Queensland, Australia, **3** Queensland Department of Natural Resources Mines and Energy, Brisbane, Queensland, Australia

* andrew.dunstan@des.qld.gov.au

**Data Availability Statement:** All relevant data are within the paper and its Supporting Information files.

## Abstract

Nester abundance is a key measure of the performance of the world's largest green turtle rookery at Raine Island, Australia, and has been estimated by mark-resight counts since 1984. Nesters are first marked by painting their carapace with a longitudinal white stripe. Painted and unpainted turtles are then counted by a surface observer on a small boat in waters adjacent to the reef. Unmanned aerial vehicles (UAV) and underwater video may provide more cost-effective and less biased alternatives to this approach, but estimates must be comparable with historical estimates. Here we compare and evaluate the three methods. We found comparatively little variation in resighting probabilities between consecutive days of sampling or time of day, which supports an underlying assumption of the method (i.e. demographic closure during sampling). This lack of bias in the location availability for detection of painted versus unpainted turtles and further supported by a parallel satellite tracking study of 40 turtles at Raine Island. Our results demonstrated that surface observers consistently reported higher proportions of marked turtles than either the UAV or underwater video method. This in turn yielded higher population estimates with UAV or underwater video compared to the historical surface observer method, which suggested correction factors of 1.53 and 1.73 respectively. We attributed this to observer search error because a white marked turtle is easier to spot than the non-marked turtle. In contrast, the UAV and underwater video methods allowed subsequent frame-by-frame review, thus reducing observer search error. UAVs were the most efficient in terms of survey time, personnel commitment and weather tolerance compared to the other methods. However, underwater video may also be a useful alternative for in-water mark-resight surveys of turtles.

## Introduction

Population abundance is a fundamental metric underpinning wildlife management that is often quantified by the capture mark-recapture survey technique, which is based on the ratio

**Funding:** The authors received no specific funding for this work.

**Competing interests:** The authors have declared that no competing interests exist.

of marked to unmarked animals in a population [1]. A less invasive and more cost-effective approach is the mark-resight approach where marked animals are subsequently visually identified without the need for physical recapture [2]. Detection of animals with naturally identifying features or artificial marks can be enhanced by technologies such as unmanned aerial vehicles (UAVs) [3], camera traps [4], passive acoustics [5] and telemetry [6]. For example, infrared camera traps have been used for mark-resight surveys of snow leopards using distinct pelage patterns [7]. Yet the limitations are ubiquitous across marine and terrestrial species, as are the assumptions that must be met. Marked and unmarked animals must be similarly detectable irrespective of environmental conditions, marks must not influence the behaviour of the animal and marks must be stable over time [1] It is therefore important to compare new methods against traditional approaches, not only in terms of the effectiveness of the approach but also in terms of whether assumptions are can be justified. This paper compares three mark-resight methods for estimating the abundance of nesting green turtles, *Chelonia mydas*: (1) a historically used surface-observer technique, (2) underwater video and unmanned aerial vehicle (UAV).

Green turtles are listed as vulnerable in the state of Queensland (*Nature Conservation Act 1992*) and in Australia (*Environment Protection and Biodiversity Conservation Act 1999*). The majority of the northern Great Barrier Reef population of green turtles nest at Raine Island (Fig 1), which is the world's largest remaining green turtle rookery [8]. Concerns about low reproductive success of green turtles at Raine Island have been reported since 1996 [9, 10], which is thought to have been caused by nesting beach inundation as well as factor related to the nest environment such as respiratory gas, microbial or temperature extremes [10]. The population is also exposed to other cumulative impacts including climate change [11], feminisation [12], hunting [13], plastic pollution [14], vessel strikes [15], commercial fishing [16] and coastal development [17]. An accurate index of nesting population numbers is critical for understanding the reproductive success and long-term changes to population numbers.

The introduction of modern technologies such as UAVs and underwater video for counting surveys coupled with artificial intelligence for automated image analysis may provide a more time efficient and reliable mark-resight estimate. Remote sensing and video techniques also permit subsequent frame-by-frame review or archiving for future analysis. Another advantage of UAVs and underwater cameras compared to the observations from a vessel is that surface reflections can be supressed or eliminated. A recent review of the uses of UAVs in marine and turtle research and management [18] identifies the broad scope of opportunities and benefits possible. The capacity to increase efficiency, reduce field personnel exposure to risks and provide new and/or better quality data gathering options, not just for population estimation, is detailed. UAVs can benefit studies on turtle nesting (including night monitoring with thermal cameras), at-sea and foraging area distribution surveys [19], a wide range of behavioural studies [18], surveillance against illegal take and hatchling dispersal and survivorship. UAVs also provide more efficient and higher resolution methods for mapping and topographic profiling of key turtle nesting and foraging habitats.

A comprehensive mark recapture program or total nesting census is unfeasible on Raine Island due to the remoteness of site coupled with the sheer number of nesters on a given night. Similarly, a total count of turtles in the inter-nesting habitat is not practical because of the limitations of detectability throughout the turtle depth range. Instead, a mark-resight approach has been used to estimate the numbers of nesters in the surrounding inter-nesting habitat since 1984 [9]. Females are painted with a white longitudinal stripe on the carapace (marked) during nightly tally counts, and counts of marked and unmarked turtles in the waters that surround Raine Island are used to estimate abundance during the sampling period using the Lincoln-Petersen estimator (LP). Mark-resight is therefore combined with in-water sampling, and thus

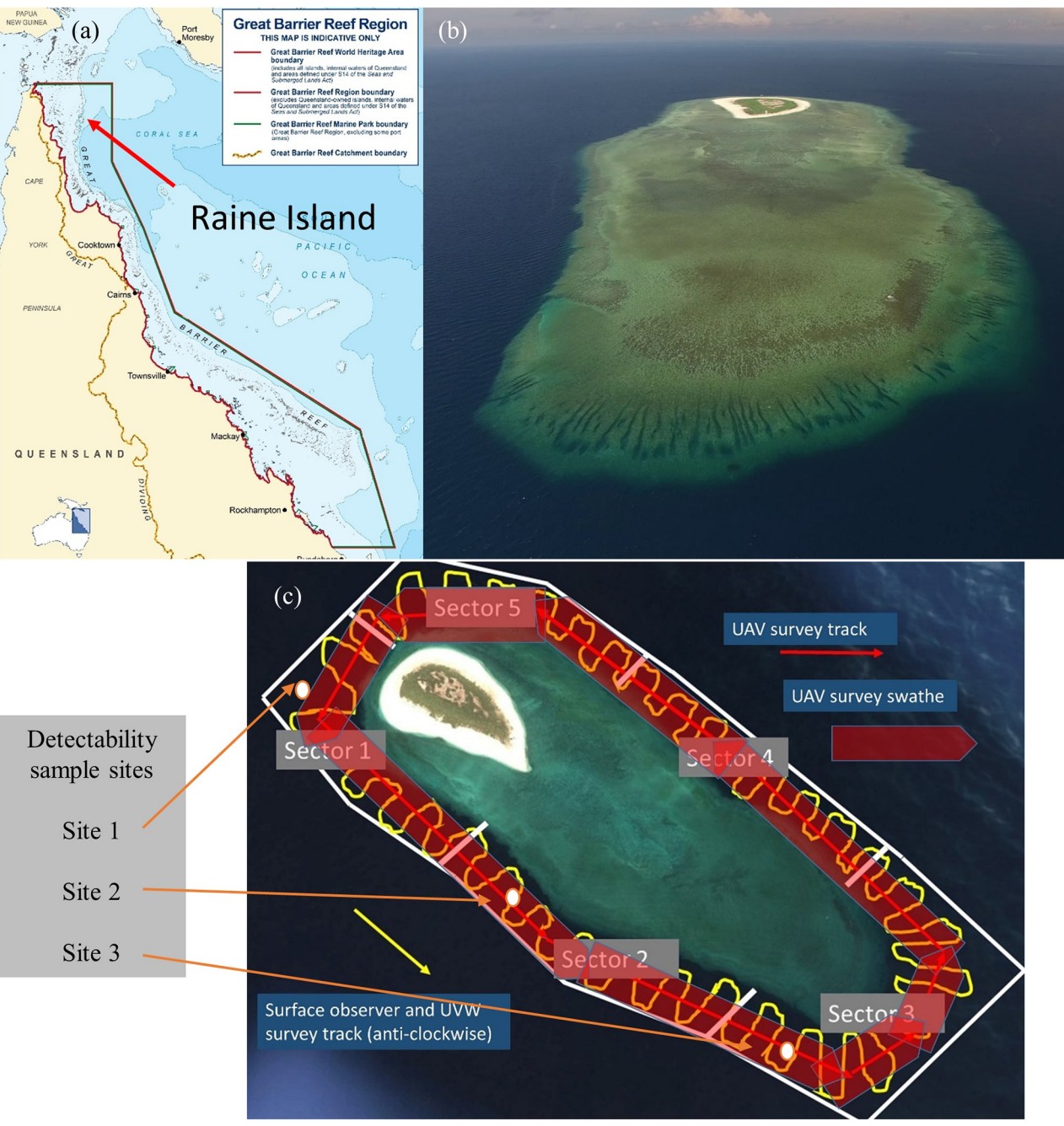

**Fig 1. Raine Island location.** (a) Location of Raine Island on the northern Great Barrier Reef, Australia, (b) Raine Island reef study site and (c) transect search paths for the three survey methods with turtle detectability experimental sample sites marked (UVW, Underwater video).

estimations of nester abundance are dependent on the limitations and assumptions of both approaches.

Here we aimed to compare the effectiveness of the vessel observers, UAVs and underwater video, and to determine if the UAV and underwater camera estimates are comparable to the historical data. The major challenge for in-water surveys is to have high detectability for both

marked and unmarked turtles, given that marine turtles spend only a small proportion of their time at the water surface, especially when surface conditions are poor, in turbid water or when turtles are amongst habitat structure [20]. The LP estimator is based on the assumption that the population is 'closed' during the sampling period [1], which means that they do not depart from inter-nesting habitat in the short time interval from marking to the in-water survey. Our comparison of detectability of marked turtles between methods also provided the opportunity to test the key LP estimator assumption of equal detectability of marked and unmarked turtles. If the probability of detection is the same between marked and unmarked turtles, the ratio of marked and unmarked turtles should not differ between sampling methods. Finally, we used a repeated sampling study design to (a) determine whether there is a gain in precision in the LP estimator with repeated sampling, and (b) test whether the closure assumption was appropriate. The LP estimator is also only based on one resighting event, which could make it less robust than estimates from repeated sampling.

## Materials and methods

### Ethics statement

All procedures used in this project were approved by the Raine Island Scientific Advisory Group and by the Queensland Department of Agriculture and Forestry Animal Ethics Committee (Permits SA 2015/12/533 and SA 2018/11/660).

### Study area

Raine Island is located on the outer edge of the northern Great Barrier Reef and is part of the Raine Island National Park (Scientific). The Wuthathi People and Kemerkemer Meriam Nation (Ugar, Mer, Erub) People are the Traditional Owners and Native Title holders for this country and are an integral partner of the area's management. Over thousands of years, Wuthathi People and Kemerkemer Meriam Nation People have held cultural connections to Raine Island through the use of its resources and cultural connections to the land and sea, through song lines, stories, and voyages to the island.

All research was undertaken on the reef waters adjacent to the Raine Island National Park (Scientific) (11˚ 35' 25" S, 144˚ 02' 05" E) between November and February during the 2013–14 to 2017–18 green turtle nesting seasons (Fig 1). Raine Island reef has a perimeter of approximately 6.5 km and is fringed by coral reefs. Green turtles are the only sea turtle species recorded nesting at Raine Island where the nesting beach is approximately 80 m wide with a circumference of 1.8 km. Nesting is seasonal with the main nesting period from October to April and extremely low rates of nesting for the rest of the year [21]. The peak nesting period is from December to January.

As many as 23,000 turtles have been counted in one night at the beach. However, there is a large variability in green turtle nesting numbers from year-to-year that is correlated with the lagged Southern Oscillation Index [22].

### Turtle marking procedure

The carapaces of nesting turtles were painted along the midline with a white stripe approximately 80 cm in length and 20 cm in width, using a 12 cm wide paint roller and "APCO-SDS fast dry water-based road marking paint" (MSDS Infosafe No. 1WDKY) [10]. A turtle was selected for painting if the carapace was dry, the carapace did not have a thick coating of algae and the turtle was inland of the beach crest (to provide sufficient time for the paint to dry). When applied under these conditions, the paint adhered to the carapace surface for at least 96

**Table 1. Summary of survey periods, number of turtles marked and survey methods conducted.**

| Survey period | Marked turtles | Number of surveys | | |
| --- | --- | --- | --- | --- |
| | | Vessel observer | Underwater video | UAV |
| Dec 2013 | 2000 | 6 | 1 | - |
| Dec 2014 | 1930 | 3 | 3 | - |
| Feb 2016 | 482 | 5 | 6 | - |
| Nov 2016 | 781 | 6 | 6 | - |
| Dec 2016 | 2000 | 6 | 5 | 3 |
| Dec 2017 | 2000 | 2 | 3 | 3 |

hr. This was confirmed over the three nights following painting of turtles, which provided the opportunity to assess the paint when many painted turtles came ashore to re-attempt nesting. While there was erosion of paint on a small proportion of turtles, enough paint always remained to allow identification of turtles as 'painted'.

Turtles were painted on a single night during turtle survey trips in November (2016), December (2013, 2014, 2016, 2017) and February (2016). All suitable turtles on the nesting beach were painted, up to a maximum of 2000 (Table 1). The upper limit was determined by logistical constraints and time while the lower limit was influenced by nesting turtle numbers.

## In-water detectability of marked and unmarked turtles

We tested the detectability of submerged green turtles using a model, which was constructed from plywood to represent an average-sized nester with curved carapace length of 106 cm [9] and was painted appropriately. The model was lowered on a rope and the depth at which it was no longer discernible as a turtle was recorded. A painted white plywood board the same size and colour as the turtle marks was then attached to the model to simulate a marked turtle. The model was again lowered to determine the depth that the white marking was still obvious. Single samples for each treatment were undertaken at three locations that represented the range of water conditions around the island from coastal aspect (site 1) to between reef channel (site 2) to open ocean aspect (site 3) (Fig 1c).

## Mark-resight counting methods

Surveys were undertaken between November and February during 2013 to 2017. Turtles were counted if the turtle shape was discernible and the presence/absence of the painted white mark was recorded. A pilot study using surface observer, underwater video and UAV methods indicated that the white markings were visibly obvious and the presence-absence of the mark was never in doubt. All unmarked turtles were considered to be adult female turtles, because previous surveys [9] demonstrated the minimal presence of adult males and juveniles during the survey period. Wind speed was mostly low during the surveys (average maximum wind speed: 11 knots, range: 1 to 18.7 knots). Water clarity measured at three sites around Raine Island using a standard Secchi disc (30 cm diameter) ranged from 9 to 13 metres.

**Surface Observer method (SO).** A standardised search area was surveyed in the waters surrounding the island on the morning and afternoon of the three days following turtle marking, or less where logistics limited sampling (Table 1 & Fig 1c). A 4.2 m outboard powered rigid hull inflatable vessel with three persons aboard, one recording, one driving and one counting, was driven along the waters adjacent to the reef perimeter edge in search of the painted turtles. The survey track (Fig 1c) followed a general pattern around the reef perimeter

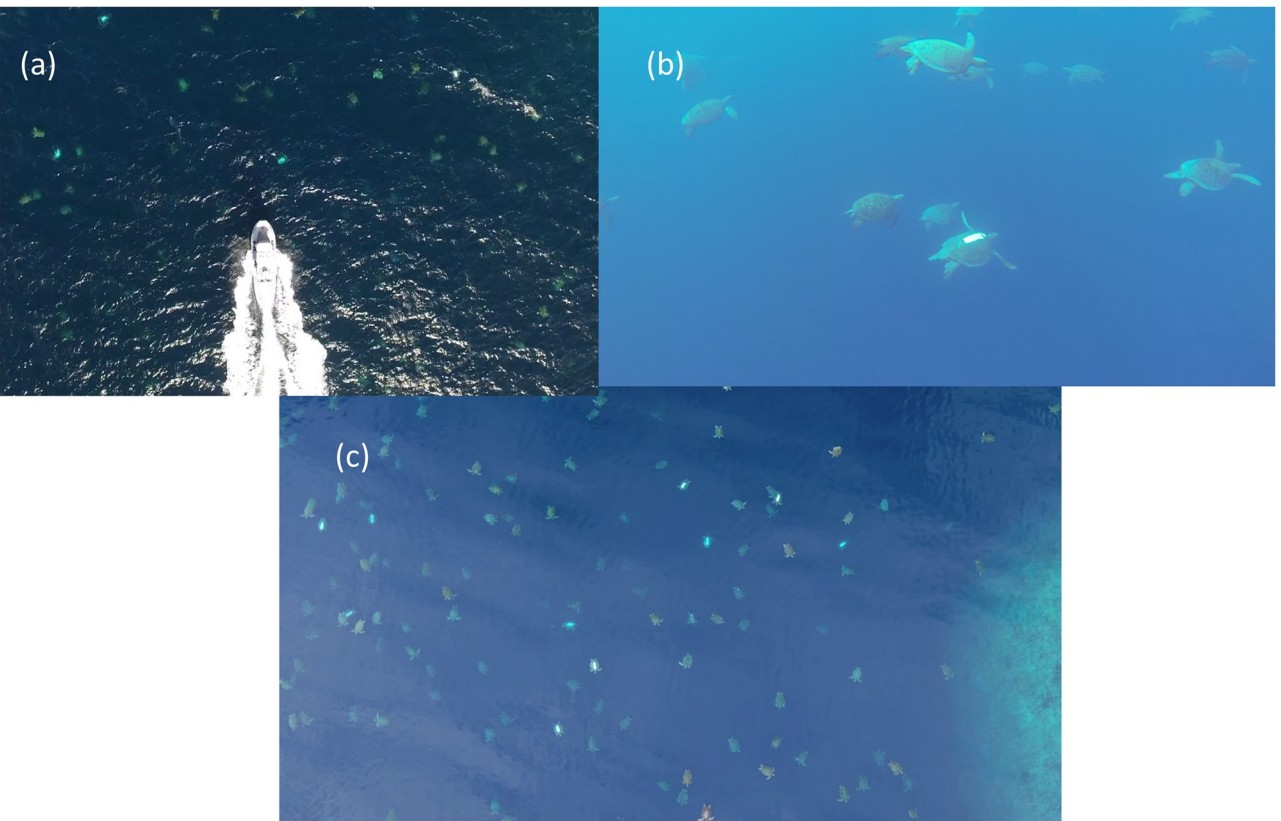

**Fig 2. Underwater video and UAV survey image examples.** (a) Still image from UAV survey in conjunction with surface observer and underwater video surveys showing survey vessel and (b) Still image from underwater video December 2017 survey and (c) still image from UAV video survey December 2017 at 50 m survey altitude.

with the vessel moving within 20 m and parallel to the reef edge for approximately 100 m then perpendicular to and away from the reef edge for 100 m, parallel to and 120 m from the reef edge for 100 m and then perpendicular to and towards the reef edge for 100 m. This pattern was repeated to complete a full circumnavigation survey of the reef. The single observer (Andrew Dunstan) was the same for all surveys. The observer was stationed at the bow of the vessel with a 180° line of sight and search area swathe of approximately 30 m as the vessel moved forward at approximately 7.4 km h$^{-1}$. This resulted in a total survey area of approximately 0.3 km$^2$.

**Underwater video method (UWV).**   Underwater video surveys were conducted from the survey vessel simultaneously with the surface observer surveys (Table 1 and Figs 1c & 2a). A GoPro Hero4 camera (frame rate: 25 hz; resolution: 1080; field of view: 127°) with an extended life battery was attached to the hull of the vessel pointing forward and downward, and recorded throughout the entire reef perimeter survey period. Underwater video visibility varied but was typically around 15 m to provide a survey swathe of approximately 60 m. Following the same track as the surface observer but with wider survey swathe resulted in a total survey area of approximately 0.4 km$^2$. Video footage was reviewed by one observer using a single tally counter to record female turtles that could be scored positively for turtle shape outline and for the presence/absence of the white paint mark during separate video replays. Video playback was paused during peak turtle density periods and playback speed adjusted for counting efficiency and accuracy.

**UAV method.**   UAV surveys were conducted as close to midday as possible to reduce sun glare on the water surface. A DJI Inspire 1 UAV with Zenmuse X3 camera (frame rate: 25 hz; resolution: 1080; field of view: 94˚; polarising filter) was flown at an altitude of 50 m and a speed of 5 m/s along a path parallel to the reef edge with the UAV visual search swathe reaching from the reef edge to 90m seawards of the reef edge (Figs 1c & 2b and S1 Multimedia). This height was selected, after trials, to provide the ability to readily identify turtles in a variety of sea conditions while providing the broadest survey swathe possible. This camera and 20 mm equivalent lens provided a horizontal video survey swathe of 90 m at the sea surface. The UAV track resulted in a total survey area of approximately 0.585km$^2$ Video footage was analysed as described for UWV surveys.

## Statistical analyses

We first compared detection depths of the turtle model (with and without the painted mark) at the three sites using a Student $t$ test on $\log_e$ transformed Secchi depths. We then compared the relative probability of detecting a marked (painted carapace) turtle between survey methods using a generalised linear mixed effects model (GLMM) with a binomial link function. A mixed-effects design was required because each batch of marked turtles was observed twice daily for five days. The optimal variance structure for the random effects was first explored using the 'lme4' package (lme4 v. 0.999375–35) of the R statistical environment (v. 2.13.1; R Development), using residual diagnostics and Akaike's Information Criterion (AIC) of different mixed models (following [23]). A model that allowed the slope of the day within nesting season effect to vary resulted in only a marginal improvement in AIC over a model that included a nested random effect of diel period (morning or afternoon) within day and nesting season. The relative probability (P) of detecting a painted green turtle (M) was therefore modelled by:

$$M_{ijk} \sim Binomial(N_{ijk}, P_{ijk})$$

$$logit(P_{ijk}) = \alpha + \beta(Method) + b_i + b_{ij} + b_{ijk} + \varepsilon_{ijk} \tag{1}$$

where the relative probability of detecting a marked green turtle (*P*) in a given time period (*i*), day (*j*) and nesting season (*k*) is a function of the survey method (*Method*). Other terms in the model are the total number of turtles that were resighted (*N*), the general intercept ($\alpha$), the random intercepts (*b*) and the residual error ($\varepsilon_{ijk}$). Eq 1 was fitted in a Bayesian framework using the 'mcmcGLMM' package and vague priors.

We then explored the gain in accuracy and precision in the LP estimator [1] from repeated recapture periods using a jacknife procedure. Each jacknife resample calculated population size as a function of the cumulative average of marked and unmarked recaptures, up to a maximum of six samples by the end of the third day (i.e. samples were taken twice daily for three days).

We estimated conversion factors for the historical estimates as the quotient of the mean population estimates, e.g. the conversion factor for surface observer to underwater video estimate was the surface observer population estimate divided by the underwater video population estimate. To explore how this conversion factor varied with population size, we fitted a linear regression of conversion factor against surface observer population size. Finally, we compared the number and densities of turtles sighted in each method using general linear models.

## Results

### In-water detectability of marked and unmarked turtles

The white mark was discernible at an average of 3 metres deeper than the unpainted turtle model ($t$ = 3.61, df = 3.8, $p$ = 0.026). The in-water detectability of painted and unpainted turtles

**Table 2. Mean values for total mature female turtles counted and Lincoln Peterson estimates for periods surveyed by each method with standard error.**

| Survey period | Vessel surface observer | | | Underwater video | | | UAV | | |
|---|---|---|---|---|---|---|---|---|---|
| | Total turtles | Peterson estimate | S.E | Total turtles | Peterson estimate | S.E | Total turtles | Peterson estimate | S.E |
| Dec 2013 | 3167.2 | **58817.8** | 6095.1 | 4289.0 | **102142.9** | 10969.9 | - | - | - |
| Dec 2014 | 1002.7 | **14439.1** | 1174.1 | 534.0 | **18827.3** | 2470.9 | - | - | - |
| Feb 2016 | 169.2 | **4708.7** | 1116.3 | 194.8 | **5398.2** | 1351.5 | - | - | - |
| Nov 2016 | 728.8 | **8838.1** | 1074.4 | 1000.5 | **13180.8** | 1756.6 | - | - | - |
| Dec 2016 | 705.5 | **12377.5** | 1089.1 | 1275.8 | **18135.9** | 1496.1 | 1460.0 | **19682.9** | 1553.2 |
| Dec 2017 | 1596.5 | **20009.4** | 1618.7 | 1679.3 | **33263.1** | 3198.1 | 4622.3 | **37035.0** | 2334.0 |

indicated that turtles were identifiable to 10 m depth, and that there were no pronounced differences in water clarity between sampling locations that were likely have influenced the results.

## Comparison of detectability between methods

Results consistently demonstrated that the proportion of marked turtles compared to all turtles sighted, was higher with the surface observer method compared to using either the underwater video or UAV. Analysis of this data translates to significantly higher LP population estimates from the UAV and underwater video methods compared to the surface observer method (Table 2 & Fig 3).

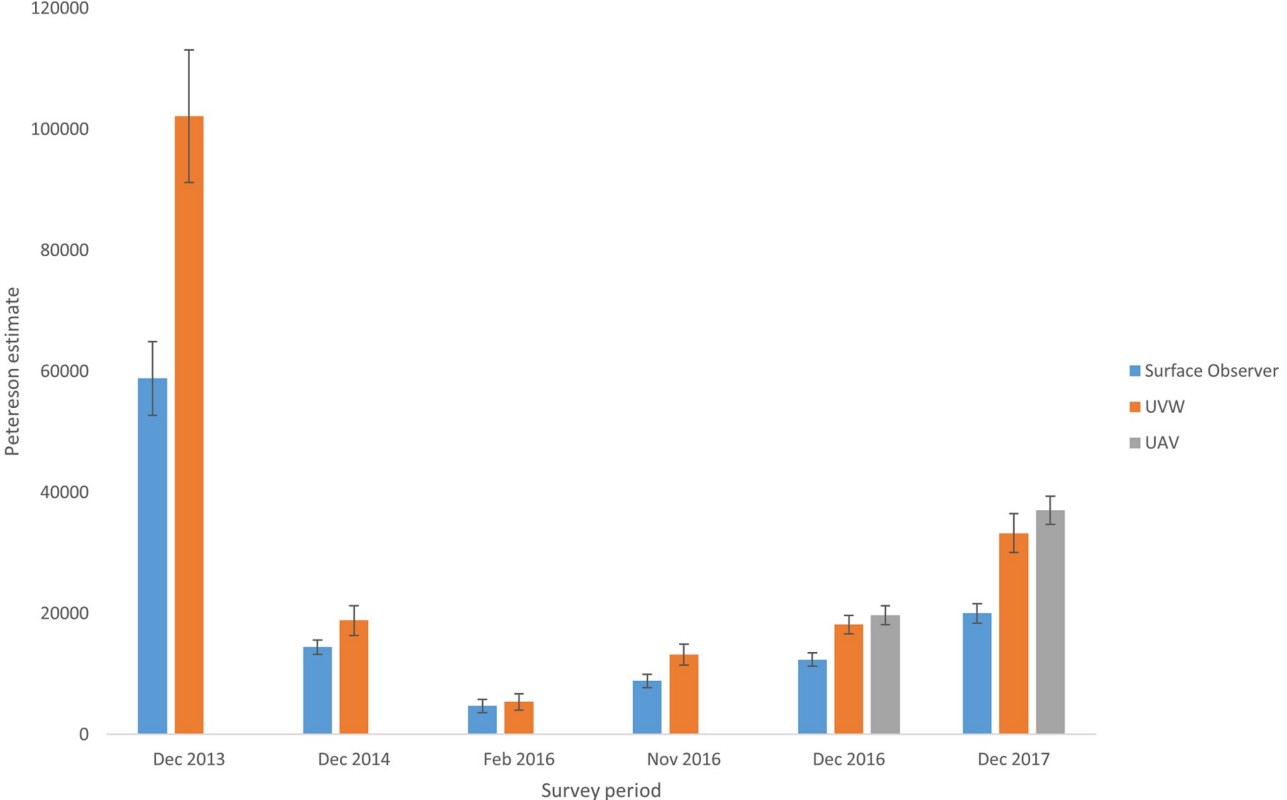

**Fig 3. Lincoln Peterson population estimates for periods surveyed by each method.** Error bars shown are ± 1 standard error. (UVW, Underwater video).

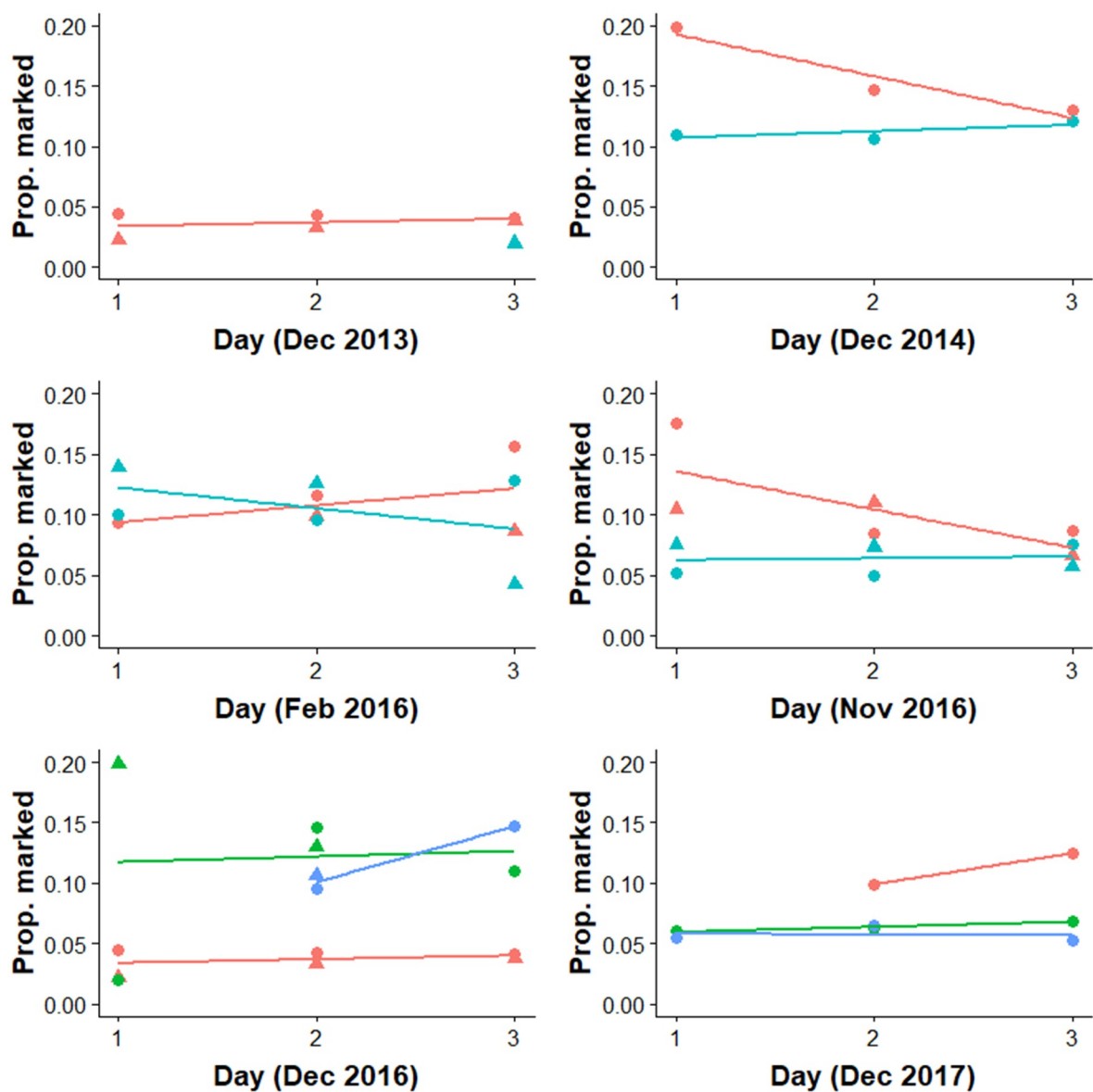

**Fig 4. Proportion of marked turtles detected for each method.** Coloured points represent the method (Red: surface observer; blue, underwater video; green, UAV) and shapes represent the diel period (circles: morning; triangles: afternoon). Samples were collected over three successive days on each occasion. The coloured lines represent the average trend over time for each method and period. Once variation associated with survey period was accounted for, there was no significant difference in detectability between the UAV and underwater video methods (Fig 5).

Survey period accounted for 96.8% of variation (highest posterior density intervals from 82.6 and 99.6%) in the relative probability of detecting a marked turtle, compared to negligible variance components associated with sampling day ($2.58 \times 10^{-5}$%, nested within sampling period) or time of day ($5.17 \times 10^{-5}$%, nested within sampling day and sampling period). On average, 9.45% of turtles detected using the surface observer method were marked (95% CI: 5.24% to 15.29%), compared to 6.58% for the underwater video method (95% CI: 3.21% to 12.02%) and 6.26% for the UAV method (95% CI: 2.86 to 12.07%) (Fig 4).

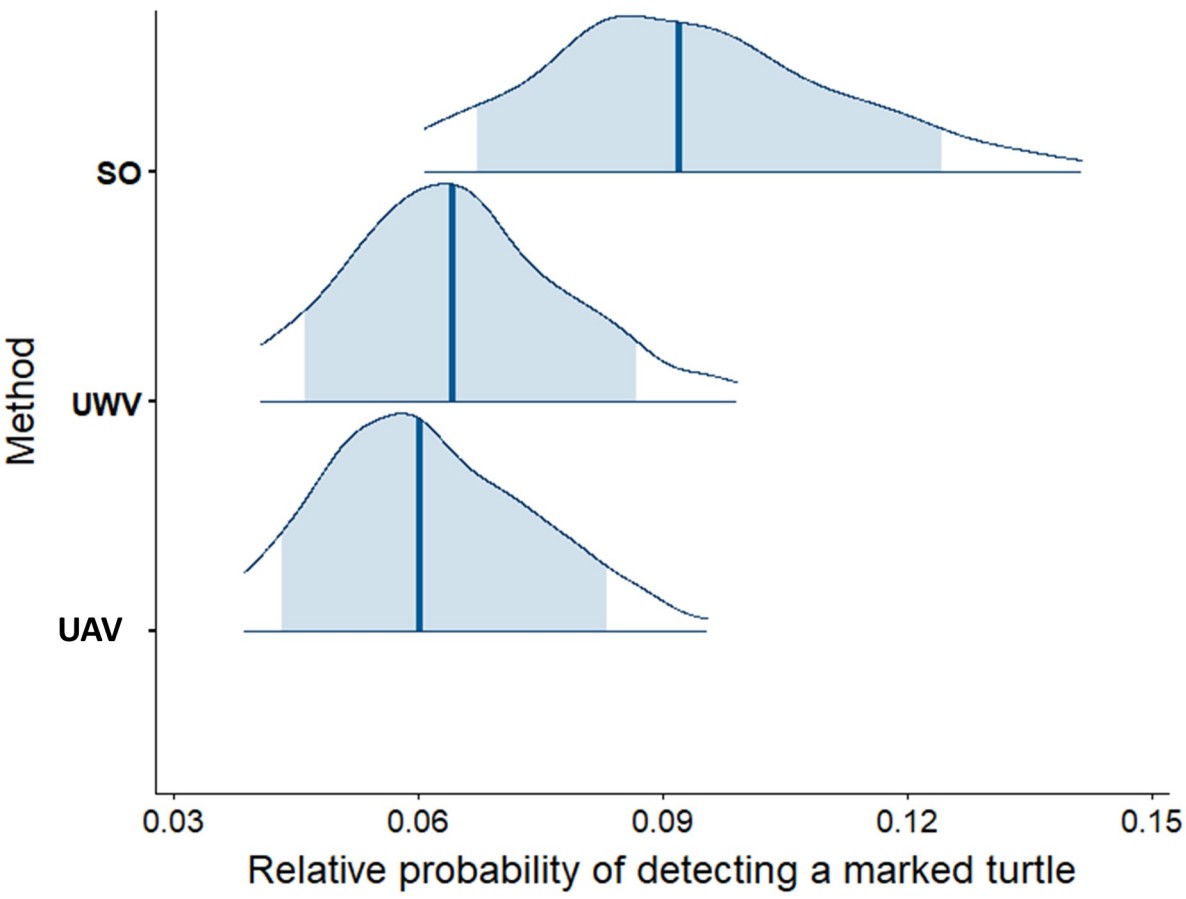

**Fig 5. Modelled relative probabilities of detecting marked turtles using each method.** (SO, surface observer; UWV, underwater video and UAV). The density plots are computed from the merged posterior draws, where the blue vertical line represents the median and shaded blue areas are the 80% credibility intervals.

The relative gain in precision from using repeated measurements was similar across all three survey methods (Fig 6). There was an obvious gain in using two measurements (rather than one). Estimates and variances stabilised after three measurements suggesting that three measurements provided adequate precision.

Due to the major differences between Lincoln-Peterson estimates using Surface observer and both underwater video and UAV techniques, the consistency of conversion factors was also investigated. Conversion factors were calculated for time periods where all three methods were undertaken at the same time, and were based on the average LP estimates. These conversion factors were then averaged to provide a mean conversion factor (Surface observer to Underwater video CF = 1.53 (SD = 0.24), Surface observer to UAV CF = 1.73 (SD = 0.18) and Underwater video to UAV CF = 1.11 (SD = 0.01)).

However, there was considerable variation in detection probabilities between sampling periods, which was likely to be driven by the extreme variability in the density of turtles in the inter-nesting habitat. In support of this suggestion, the Underwater video:Surface observer correction factor tended to increase with the population estimate (Fig 7), however this relationship was not significant likely because of the small sample size (n = 6, spearman-rank correlation test rho = 0.69, p = 0.125).

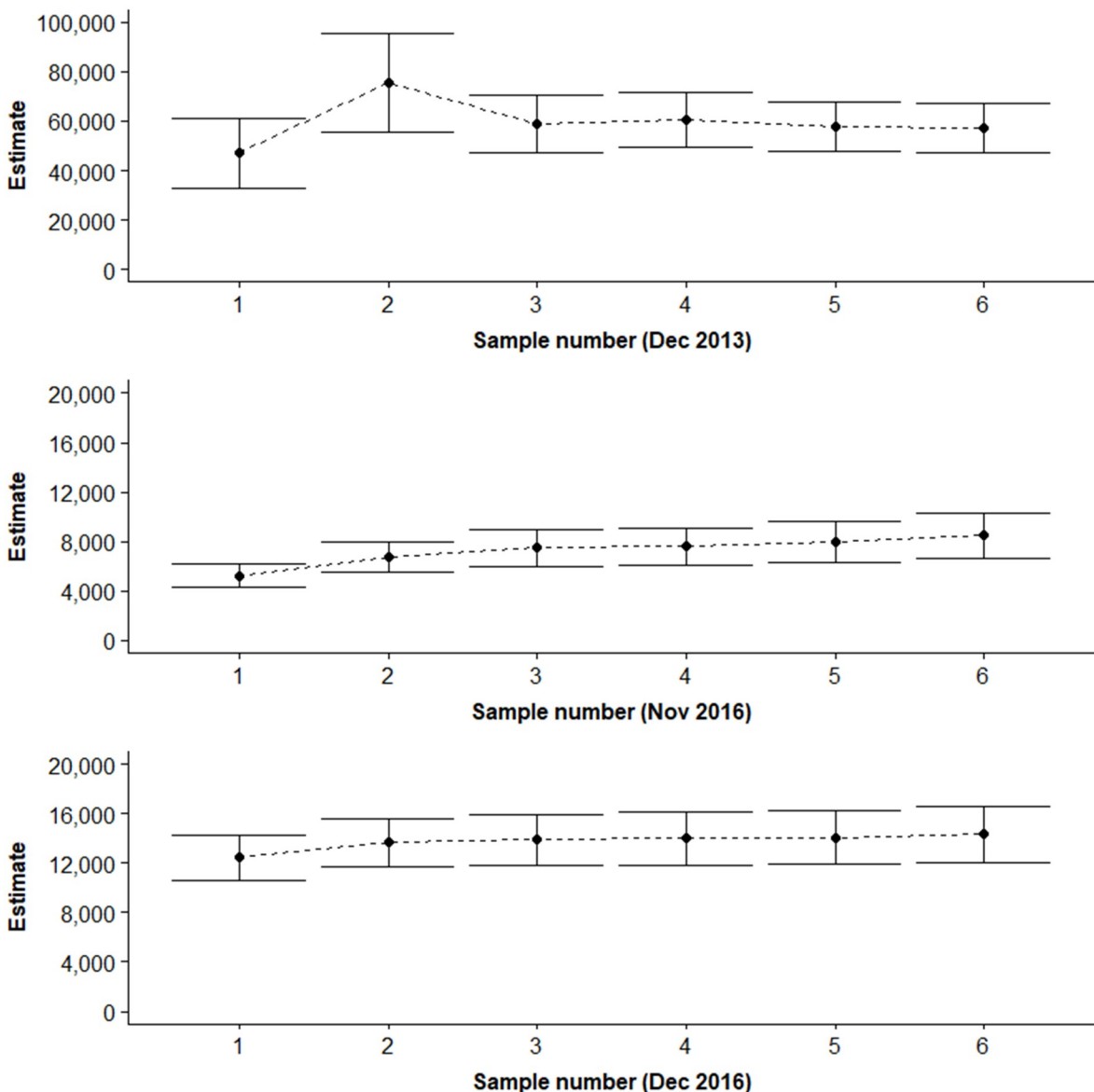

**Fig 6. Influence of sample size on the Lincoln-Petersen estimate.** Shown here are the estimates for the Surface observer method and the three sampling periods for which six samples were available (± 95% confidence intervals).

The conversion factors were applied to historical data collected by the surface observer method to provide a complete time series of Raine Island breeding female population numbers during early December since 1987 (Fig 8). There are a number of seasons where surveys were not conducted and these data points are missing.

The use of UAVs to conduct mark/resight surveys is considerably more efficient in survey time (1:2.5 hrs) and personnel commitment (1:3) than the other survey techniques. UAV surveys can also be conducted in more extreme weather conditions (13:8 ms$^{-1}$) while still providing precise estimates (Table 3). Consistent rain negates UAV flight options but is not a major impact on the other methods.

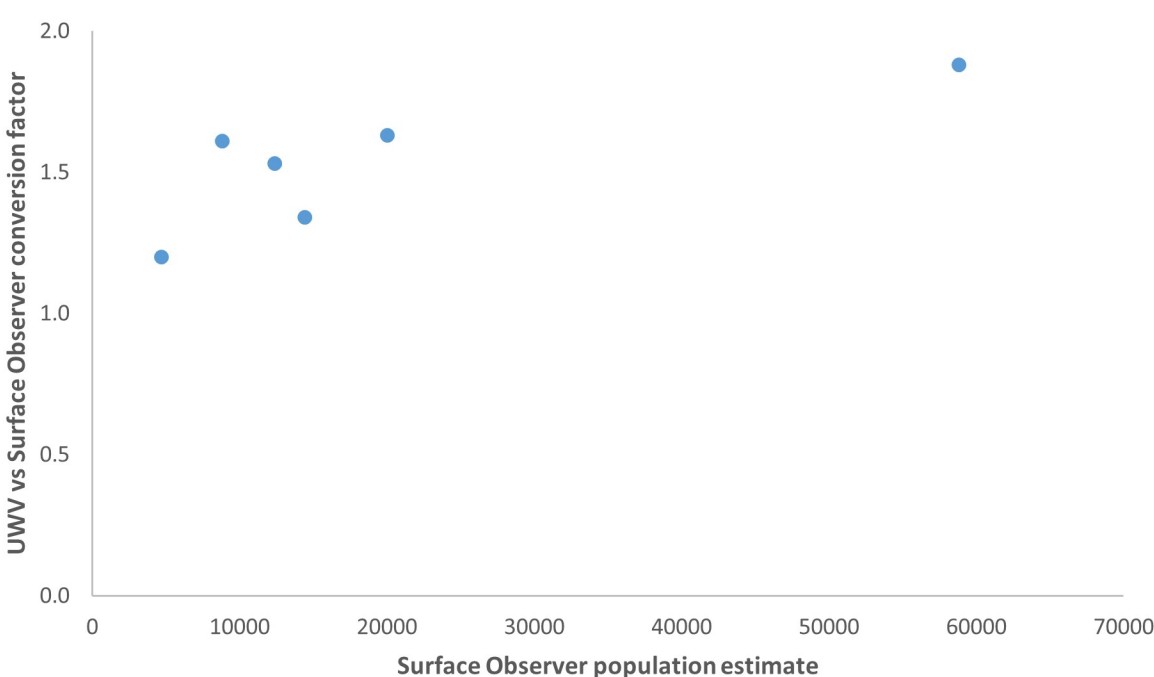

**Fig 7. Conversion factor ratio of underwater video vs surface observer methods compared with population estimates from surface observer surveys from different seasons.** (UVW, Underwater video).

UAVs also searched a larger search swath than the other two methods, resulting in 0.585 km$^2$ searched on each occasion, compared to an estimated 0.4 km$^2$ for the underwater video method (assuming a distance of 15m and a viewing angler of 127°) and 0.3 km$^2$ for the surface observer method (assuming a search radius of 15m from the vessel). The average number of

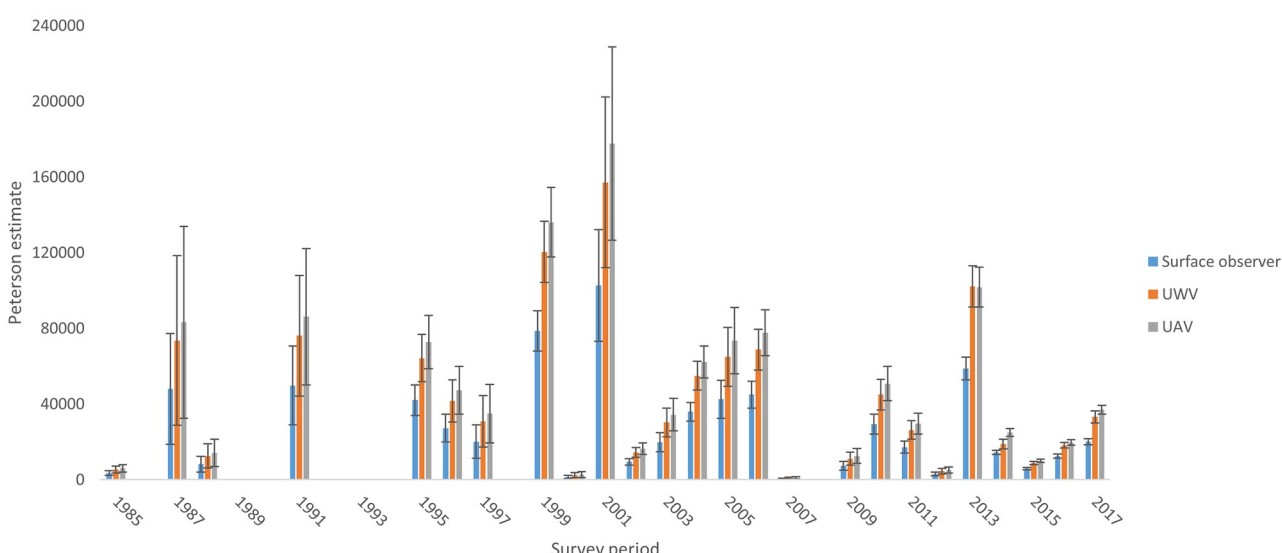

**Fig 8. Historical population estimates for December sampling periods using surface observer method presented with conversion factors applied for underwater video and UAV equivalent estimates.**

**Table 3. Comparison of the cost effectiveness and logistical considerations for each turtle count method.**

| Survey method | Equipment Cost | Personnel | Survey time | Viable wind conditions |
|---|---|---|---|---|
| Surface observer | Low | 3 | 2.5 hrs | 8 ms$^{-1}$ |
| Underwater video | Low-moderate | 3 | 2.5 hrs | 8 ms$^{-1}$ |
| UAV | Moderate | 1 | 1 hr | 13 ms$^{-1}$ |

*The survey time period refers to the total time to cover transects as shown on Fig 1.

turtles counted by the UAV tended to be higher (3041) than the other methods (Surface observer: 1228; Underwater video: 1345) (Table 2 and Fig 9). However, neither total numbers nor densities significantly differed between methods (log$_e$ total numbers: D = 2.737, df = 2, p = 0.375; log$_e$ density: D = 0.458; df = 2, p = 0.795).

## Discussion

The UAV and underwater video methods detected a lower ratio of marked to unmarked turtles than the surface observer method, resulting in considerably higher estimates of nester

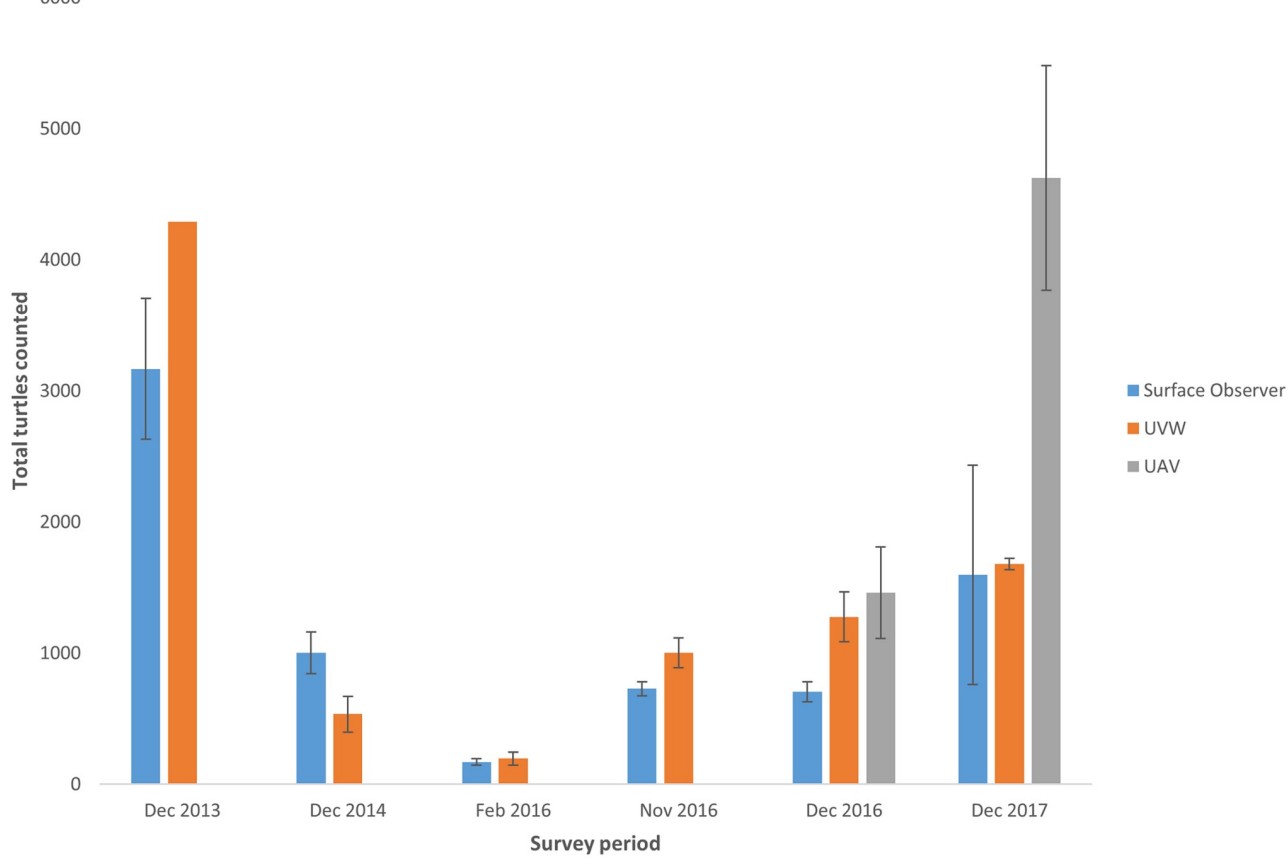

**Fig 9. Mean total turtles counted (painted + unpainted) for periods surveyed by each method.** Error bars shown are ± 1 standard error. (UVW, Underwater video).

abundance. UAVs yielded a population estimate 1.73x higher than the historical surface observer method, whereas the underwater video method estimated 1.53x more turtles than the surface observer method. However, there was considerable variation in detection probabilities between sampling periods, which was likely to be driven by considerable variability in the density of turtles in the inter-nesting habitat, suggesting that robust correction factors would require more sampling across a range of turtle densities. The oceanic waters surrounding Raine Island are extremely clear with visibility greater than 10m and often more than 20 m. This may impose limitations for adapting this methodology to locations with lower visibility especially inshore habitats.

A key advantage of the underwater video and UAV approaches is the ability to review and playback video at speeds most suitable for accurate counts, especially when turtle aggregations were dense. The biased attraction of painted turtles to the observer's eye is not tested or quantified but is considered to be the major factor causing the higher percentage of painted turtles recorded by the surface observer, resulting in lower overall population estimates by this method. We posit that the marked differences in the detection rate of marked turtles between the photographic and visual observer methods is due to visual searching limits of human observers. The performance of visual searching, as measured by search accuracy or reaction time, typically declines as the number of objects increases [24, 25]. Observer fatigue can also influence detection rates [26]. Further, in analysing complex natural or visually noisy scenes, humans direct visual attention towards regions of high contrast attract visual attention, particularly reflective surfaces such as white paint that represent high luminance contrast [27]. This effect would be even greater in the noisy environment caused by surface reflections or surface disturbance [26]. In our experiment, the white mark was discernible three meters deeper than the turtle model, suggesting that it may have drawn the attention of an observer who was subsequently able to discern that it was a turtle. Together, these mechanisms may explain why a visual observer had a higher probability of identifying marked turtles than the UAV or underwater video approach. This may also explain the fact that detection probability was the most similar between the underwater video and the visual observer in February 2016, when the population estimate was the lowest (Figs 3 & 8 and Table 2). We predict that search accuracy would be greater when there are fewer turtles.

The in-water detectability of painted and unpainted turtles indicated that turtles were identifiable to 10 m depth from UAV height of 50m, and that there were no pronounced differences in water clarity between sampling locations that were likely have influenced the results. These results are similar to those previously reported [28] where UAV flights at 60m were the maximum height for reliable detection of turtles in an inshore habitat to a depth of 7m at the seabed. However, we did not test how the viewing angle and surface conditions influenced detectability. Counting from the surface observer platform was mostly conducted at an angle to the surface of the water, and hence more subject to interference from glare and surface disturbance than the UAV or underwater video method. This may have also influenced the ratio of painted to unpainted turtles detected, because the paint remains visible during these conditions.

Compared to variation between sampling periods, there was little variation association with the timing of sampling or over consecutive samples. This suggests that the population is closed during sampling, an assumption also supported by the results of two other parallel studies. Firstly, the rate of mortality is low, with a maximum of 0.045% during the sampling period (interpolated from Robertson et al., in prep). Secondly, recently satellite tracking of 40 nesters at Raine Island in the 2017–18 and 2018–19 nesting seasons indicated that the vast majority of turtles remained in the immediate vicinity of the Raine Island reef edge after successful or unsuccessful laying. This study also supported a lack of bias in the location availability for

detection of painted versus unpainted turtles. It demonstrated no significant difference between presence within the survey area during the first three days post nesting (the survey period) and the remaining internesting period (Mark Hamann, James Cook University, pers. comm.).

The use of UAVs to conduct mark-resight surveys is considerably more efficient in survey time (1:2.5 hrs) and personnel commitment (1:3) than the other survey techniques. Video analysis to count turtles is done manually at present however automated image analysis techniques are almost complete and will remove this extra time and personnel requirement. UAV surveys also still provide quality data when the sea-surface state and wind (i.e. 8–13 ms$^{-1}$ winds) limit the surface observer or underwater video methods, although consistent rain hinders the use of UAVs. The efficiency of the UAV method also facilitates cost-effective optimisation of the study design by using resampling to increase the precision of the population estimates (Fig 6).

Methods for manually counting wildlife from remotely sensed imagery are well documented [29]. In our study, the use of video recording versus the use of overlapping still images to produce a single orthomosaic image by UAV were both considered. For this application the benefit of moving video images during counting review provided the ability to adjust playback and pause footage to enable each individual turtle to be assessed as the UAV moved past. Movement of individual turtles was then used as part of authenticating turtle recognition, to gain different angle and reflectance aspects to optimise clarity of each turtle and paint mark and to allow the closest point of contact to be used in assessment (S1 Table).

Although no other studies have used UAVs in conjunction with mark-resight to estimate turtle abundance [30], other turtle UAV studies have used the direct count method for in-water [28, 19] and operational sex ratios [28]. Aerial surveys for nesting beach track counts [31] may be more effectively undertaken by UAVs in the future. In-water UAV abundance counts of turtles are adjusted for the availability bias [30], however these adjustments were not deemed necessary in the Raine Island study due to the very clear waters that allowed detection to at least 10 m in water depth. The proportion of time spent by turtles in this 10 m detectable range is currently being investigated through studies of time depth recorders deployed on 21 nesters at Raine Island during the 2018–19 season. This will inform any bias of detectability for this mark-resight study and for use in total turtle counts conducted in other locations. Even acknowledging these limitations, our total and density estimates using the UAV survey method are higher than UAV density measurements of olive ridley turtles (*Lepidochelys olivacea*) at Costa Rica, the only other mass sea turtle nesting aggregation in the world. During the low-medium level nesting season in 2016 and the medium level nesting season in 2017 densities were 2496 ± 1441 turtles · km$^{-2}$ and 7901 ± 1465 km$^{-2}$ respectively. Low and high-end estimates of turtle density at Costa Rica were 1299 ± 458 km$^{-2}$ and 2086 ± 803 km$^{-2}$ respectively [19].

## Conclusions

In summary, this study indicates that the use of UAVs for in-water mark-resight turtle population estimation is an efficient and accurate method that can provide an accurate adjustment for historical adult female population estimates at Raine Island. Underwater video may continue to be used as a backup method in case of UAV failure or weather restrictions to flight. This study also provides the basis for accurate nesting population estimation, including historical data correction, to inform reproductive success parameters for green turtles at Raine Island. This knowledge is crucial to identify the causes and quantify the levels of nesting and hatching failure and hatchling production. The data is also essential to the evaluation of

improvements in reproductive success resulting from conservation management interventions such as re-profiling of the nesting beach and fencing to reduce adult female mortality [10].

## Supporting information

**S1 Multimedia. UAV survey video.** Video of UAV survey at 50m altitude over waters adjacent to Raine Island reef edge.
(MP4)

**S1 Table. Survey area and estimated densities of turtles sighted using each method.**
(DOCX)

**S1 Raw data.**
(XLSX)

## Acknowledgments

The Raine Island Recovery Project is a five-year collaboration between BHP, the Queensland Government, the Great Barrier Reef Marine Park Authority, Wuthathi and Meriam Nation (Ugar, Mer, Erub) Traditional Owners and the Great Barrier Reef Foundation to protect and restore the island's critical habitat to ensure the future of key marine species.

## Author Contributions

**Conceptualization:** Andrew Dunstan, Richard Fitzpatrick.

**Data curation:** Andrew Dunstan, Katharine Robertson, Jeffrey Pickford.

**Formal analysis:** Jeffrey Pickford, Justin Meager.

**Investigation:** Andrew Dunstan, Katharine Robertson, Richard Fitzpatrick.

**Methodology:** Andrew Dunstan, Richard Fitzpatrick, Jeffrey Pickford.

**Project administration:** Andrew Dunstan.

**Resources:** Andrew Dunstan.

**Software:** Andrew Dunstan, Richard Fitzpatrick.

**Supervision:** Andrew Dunstan.

**Validation:** Andrew Dunstan, Justin Meager.

**Visualization:** Andrew Dunstan.

**Writing – original draft:** Andrew Dunstan, Justin Meager.

**Writing – review & editing:** Justin Meager.

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
