## [Decision Letter · Decision Letter 0]

11 Feb 2020

PONE-D-20-01406

Use of unmanned aerial vehicles (UAVs) for mark-resight nesting population estimation of adult female green sea turtles at Raine Island

PLOS ONE

Dear Andrew,

Thank you for submitting your manuscript to PLOS ONE. After careful consideration, we feel that it has merit but does not fully meet PLOS ONE’s publication criteria as it currently stands. Therefore, we invite you to submit a revised version of the manuscript that addresses the points raised during the review process.

Andrew - you will see constructive referee comments that, with a little care, will be straightforward to address. I look forward to seeing a revision. Cheers, Graeme

We would appreciate receiving your revised manuscript by Mar 27 2020 11:59PM. To enhance the reproducibility of your results, we recommend that if applicable you deposit your laboratory protocols in protocols.io, where a protocol can be assigned its own identifier (DOI) such that it can be cited independently in the future. For instructions see: http://journals.plos.org/plosone/s/submission-guidelines#loc-laboratory-protocols

We look forward to receiving your revised manuscript.

Kind regards,

Graeme Hays

Academic Editor

PLOS ONE

Additional Editor Comments (if provided):

Dear Andrew,

We now have two detailed and constructive reviews of your manuscript. You will see that the referees generally like the manuscript but there are a few suggestions. I think that if you take care with a revision, this manuscript will be still be suitable for PLoS 1 and will make a fine paper, so I am asking that you to revise the manuscript taking the comments into consideration.

In addition to the referee comments, I have a few suggestions from my own reading:

1. Line 16. “We hypothesise that the UAV and UWV methods improved detection rates of marked turtles …”. I could not understand. Detection rates of marked turtles were lower using the UAV and UWV compared to the SO method. Do mean detection rates of unmarked turtles ?

2. Line 210. You say “Results consistently demonstrated a higher detection ratio of marked:unmarked turtles using UAV and UWV when compared with the SO method.”

But then a few lines later when you report the differences, you present the results in a different way:

Line 226. “On average, 9.45 % of turtles detected using the SO method were marked (95% CI: 227 5.24% to 15.29%), compared to 6.58% for the UWV method …”

To avoid confusion, I would present results the same way, i.e. start the Results with a statement along the lines of: “Results consistently demonstrated that the % of marked turtles compared to all turtles sighted, was higher with the SO method compared to using either the UAV or UWV.”

3. Fig. 3. Are you able to splice you estimated numbers with the previous published time-series for Raine Island, to show the current status compared to previous estimates ?

I look forward to seeing a revision.

All best wishes, Graeme Hays

Journal Requirements:

2. We note that Figure 1 in your submission contains map images which may be copyrighted. All PLOS content is published under the Creative Commons Attribution License (CC BY 4.0), which means that the manuscript, images, and Supporting Information files will be freely available online, and any third party is permitted to access, download, copy, distribute, and use these materials in any way, even commercially, with proper attribution. For these reasons, we cannot publish previously copyrighted maps or satellite images created using proprietary data, such as Google software (Google Maps, Street View, and Earth). For more information, see our copyright guidelines: http://journals.plos.org/plosone/s/licenses-and-copyright.

"The Raine Island Recovery Project is a five-year, $7.95 million collaboration between

BHP, the Queensland Government, the Great Barrier Reef Marine Park Authority, Wuthathi

and Meriam Nation (Ugar, Mer, Erub) Traditional Owners and the Great Barrier Reef

Foundation to protect and restore the island’s critical habitat to ensure the future of key

marine species.

The funders had no role in study design, data collection and analysis, decision to publish,

or preparation of the manuscript."

"The authors received no specific funding for this work"

4. Please amend the manuscript submission data (via Edit Submission) to include author Justin Meager.

Reviewers' comments:

Reviewer's Responses to Questions

**Comments to the Author**

1. Is the manuscript technically sound, and do the data support the conclusions?

Reviewer #1: Yes

Reviewer #2: Yes

2. Has the statistical analysis been performed appropriately and rigorously? 

Reviewer #1: Yes

Reviewer #2: Yes

3. Have the authors made all data underlying the findings in their manuscript fully available?

Reviewer #1: No

Reviewer #2: Yes

4. Is the manuscript presented in an intelligible fashion and written in standard English?

Reviewer #1: Yes

Reviewer #2: Yes

5. Review Comments to the Author

Reviewer #1: Overview

The overall objective was very interesting, in comparing the three methods for detecting turtles that are already marked to maintain consistent monitoring effort of the population. I have suggested reorganising how you present the results, and clarifying that the surface observer result is an artefact of human bias, as this is not clear until the discussion, where it is beautifully presented. Many of the issues addressed are of broad interest to a general audience, however, the organisation of the Introduction limits this. I have suggested reorganizing how the information in this section is presented. I have advised adding more details to the methods to help clarify the different features of the approaches, while obvious to you, a general reader will not be able to perceive the different limits of each approach. I have suggested drawing on existing studies of sea turtles and other marine animals where some of the criteria delineated or described in methods and discussion have actually been addressed. Overall, this was a very interesting study, and a pleasure to read. Once these minor changes have been made, I believe this will make a very useful contribution to the journal.

Abstract

I would advise avoiding using the abbreviations for the three approaches, it is very hard to follow the context for general readers.

Please clarify in an abstract what the marking is – this would impact detectability by the different methods.

Line 8 versus 16: I believe these results need to be reordered, 1, the detection rates and 2, the relative likelihood of resighting. Here, I would advise stressing that this is because from the surface a white marked turtle is more easily spotted than a non white marked turtle, making this approach biased to observer ability.

Please check formatting specifications; I think PlosOne specify a single paragraph format.

Introduction

PlosOne draws on a wide audience. I would advise starting the manuscript with the broad issue, i.e. the use of different methods to gauge population abundance – this is generic across marine and terrestrial species, with the issues of different technologies being ubiquitous.

Line 48, this is more methodological, In any case, clarify what is painted – numbers/codes or just colours? I would move this to the end of the introduction or start of methods.

Line 54, I would draw on this approach in a more general context and why it is better/more relevant than other approaches, identifying possible alternatives used on other wildlife. – i.e. I would make this my second paragraph, but more generic

Line 64, good paragraph, this should be your second or third depending on how you plan to organise it, i.e. advances in data collection approaches followed by how to analyse it seems more logical…

Line 68, here start a new paragraph.

Methods

Line 142, More details are required here. Please clarify the line of sight, number of observers and whether all 360 degrees was surveyed continuously. Please state whether transect lines were perpendicular or parallel to shore, the distance between transects and the total area covered.

Line 148, please state what the underwater field of view is, i.e. generally visibility is less compared to air, due to particles in the water.

Line 164, why was this height selected? IN the figure, it shows that you flew it parallel to shore, how far offshore was the flight path? I would advise delineating this on the figure. Not jus the path line, but the swathe.

Phenomenal effort to collect data using all three approaches in tandem.

The white stripe makes the turtles extremely conspicuous; as this approach has been used for many years, is there a greater predation risk of these turtles compared to unmarked turtles? If you know the ratio of marked to unmarked turtles in the water and the ratio of marked to unmarked turtles with trauma/death, this would give some quantitative value. Also, with the drones, you might be able to view whether potential predators aggregate around marked turtles more.

Results

Line 205 – this should be just results, not results and discussion, as you present the discussion later.

Line 206 - It would be useful/interesting to clarify for each approach, i.e. boat based observation – what was the greatest depth and distance of turtles observed by observers; underwater observation – same again, what was the greatest distance underwater and how deep; drones – same again, how deep could you detect turtles – were any on the seabed allowing you to determine this?

This helps a reader understand the relative potential of each approach.

Line 277, good point, and nicely presented

Line 279, agreed – also providing insights under conditions that would not be possible in a boat.

Line 287, yes, but more detail is needed in the methods on the relative fields of view of the different methods to support this.

Line 304, here, you should compare this result to that obtained by Schofield et al. It is interesting that the detection rates were not even higher for the drone:

Schofield G, Katselidis KA, Lilley MKS, Reina R, Hays G. 2017. Detecting elusive aspects of wildlife ecology using UAVs: new insights on the mating dynamics and operational sex ratios of sea turtles. Functional Ecology 31 (12), 2310–2319 DOI: 10.1111/1365-2435.12930

Hodgson, A., Kelly, N., Peel, D. 2013. Unmanned aerial vehicles (UAVs) for surveying marine fauna: a dugong case study. PLOS ONE 8: e7955

Line 305, this implies that the underwater approach obtained more turtles than the drone? Is there any explanation of why? Is the underwater visibility exceedingly good at your study location. I know that at other locations, for it to exceed 5 m is rare. This should be stressed that this result might differ to other locations where underwater visibility is not so clear.

Line 311, yes, absolutely. The observer would also start only looking for white strips due to it being easier. This needs to be clarified in the Abstract too, because, at present the way it is presented implies a robust/objective output, rather than observer issues.

Line 316, absolutely.

Line 321, yes, very good observation/reference.

Line 331, it would be good to present this in methods/results for all three approaches.

Line 351, this should be your first key finding presented in the abstract.

Line 351, paragraph – there are several studies on monitoring sea turtles with drones that have been published and should be referred to/compared here. See the following paper for a review of the various papers on which to draw information:

Schofield G, Esteban N, Katselidis KA, Hays GC. 2019. Drones for research on sea turtles and other marine vertebrates – A review. Biological Conservation https://doi.org/10.1016/j.biocon.2019.108214

Bevan, E., Wibbels, T., Najera, B.M., Martinez, M.A. et al., 2015. Unmanned aerial vehicles (UAVs) for monitoring sea turtles in near-shore waters. Mar. Turtle Newsl. 145, 19−22

Line 364 – these aspects have been justified by previous studies, so it is worth drawing on these to support these decisions.

Line 372, again, see the following paper where the potential detectability of turtles at different seabed depths was investigated

Schofield G, Katselidis KA, Lilley MKS, Reina R, Hays G. 2017. Detecting elusive aspects of wildlife ecology using UAVs: new insights on the mating dynamics and operational sex ratios of sea turtles. Functional Ecology 31 (12), 2310–2319 DOI: 10.1111/1365-2435.12930

Figure 2 – I would advise adding a photo of boat based observation too to present all three approaches here.

Reviewer #2: A few minor comments to improve this nice, concise manuscript.

- In the introduction, I suggest that some more context for use of UAVs is provided, e.g. the sorts of uses outlined in Rees et al 2018 The potential of unmanned aerial systems for sea turtle research and conservation: a review and future directions. Endangered Species Research 35: 81-100.

- analysis of UAV and UWV video: was there any analysis done of learning effect to account for the viewer becoming more expert at detecting turtles as more video was observed?

- was a polarising or UV filter used on the UAV camera?

- delete 'on' line 338

- author name is missing in reference 23

- general comment for the discussion: is there opportunity for automated video analysis by image search of UAV and UWV video by computer?

- notwithstanding the assumption that they are absent, were males ever observed by any of the survey techniques?

6. PLOS authors have the option to publish the peer review history of their article (what does this mean?). If published, this will include your full peer review and any attached files.

Reviewer #1: No

Reviewer #2: Yes: Richard Reina

---

## [Author Response · Author response to Decision Letter 0]

7 Apr 2020

Response to Reviewers

PONE-D-20-01406

Use of unmanned aerial vehicles (UAVs) for mark-resight nesting population estimation of adult female green sea turtles at Raine Island

PLOS ONE

Attn: Graeme Hays

Editor PlosONE

Dear Graeme,

I thank you and the reviewers for your constructive comments and suggestions and have detailed my changes and responses below. I hope these revisions address all the points appropriately and render the submission ready for publication.

Kind regards

Andrew Dunstan

Additional Editor Comments ():

In addition to the referee comments, I have a few suggestions from my own reading:

1. Line 16. “We hypothesise that the UAV and UWV methods improved detection rates of marked turtles …”. I could not understand. Detection rates of marked turtles were lower using the UAV and UWV compared to the SO method. Do mean detection rates of unmarked turtles ?

Response: Yes, this is correct and has been changed to make the message clearer 

Line 13

Our results also demonstrated that surface observers consistently reported higher proportions of marked turtles than either the UAV or underwater video method. This in turn yielded higher population estimates with UAV or underwater video compared to the historical surface observer method, which suggested correction factors of 1.53 and 1.73 respectively. We attributed this to observer search error because a white marked turtle is easier to spot than the non-marked turtle. In contrast, the UAV and underwater video methods allowed subsequent frame-by-frame review, thus reducing observer search error. 

2. Line 210. You say “Results consistently demonstrated a higher detection ratio of marked:unmarked turtles using UAV and UWV when compared with the SO method.”

But then a few lines later when you report the differences, you present the results in a different way:

Line 226. “On average, 9.45 % of turtles detected using the SO method were marked (95% CI: 227 5.24% to 15.29%), compared to 6.58% for the UWV method …”

To avoid confusion, I would present results the same way, i.e. start the Results with a statement along the lines of: “Results consistently demonstrated that the % of marked turtles compared to all turtles sighted, was higher with the SO method compared to using either the UAV or UWV.”

Yes, revised as suggested

Line 258

Results consistently demonstrated that the proportion of marked turtles compared to all turtles sighted, was higher with the surface observer method compared to using either the underwater video or UAV. Analysis of this data translates to significantly higher LP population estimates from the UAV and underwater video methods compared to the surface observer method (Table 2 & Fig 3)

3. Fig. 3. Are you able to splice your estimated numbers with the previous published time-series for Raine Island, to show the current status compared to previous estimates ?

Done, and included as Fig8. 

I look forward to seeing a revision.

All best wishes, Graeme Hays

Journal Requirements:

Response: reformatted as requested

2. We note that Figure 1 in your submission contains map images which may be copyrighted. All PLOS content is published under the Creative Commons Attribution License (CC BY 4.0), which means that the manuscript, images, and Supporting Information files will be freely available online, and any third party is permitted to access, download, copy, distribute, and use these materials in any way, even commercially, with proper attribution. For these reasons, we cannot publish previously copyrighted maps or satellite images created using proprietary data, such as Google software (Google Maps, Street View, and Earth). For more information, see our copyright guidelines: http://journals.plos.org/plosone/s/licenses-and-copyright.

Response: 

replaced with GBRMPA copyright image – an internally copyrighted image from the author’s partner organisation.

"The Raine Island Recovery Project is a five-year, $7.95 million collaboration between

BHP, the Queensland Government, the Great Barrier Reef Marine Park Authority, Wuthathi

and Meriam Nation (Ugar, Mer, Erub) Traditional Owners and the Great Barrier Reef

Foundation to protect and restore the island’s critical habitat to ensure the future of key

marine species.

The funders had no role in study design, data collection and analysis, decision to publish,

or preparation of the manuscript."

"The authors received no specific funding for this work"

Statement removed from acknowledgements section and appropriate wording added to Funding statement

4. Please amend the manuscript submission data (via Edit Submission) to include author Justin Meager.

Amended on resubmission 

Reviewers' comments:

Reviewer's Responses to Questions

Comments to the Author

1. Is the manuscript technically sound, and do the data support the conclusions?

Reviewer #1: Yes

Reviewer #2: Yes

2. Has the statistical analysis been performed appropriately and rigorously? 

Reviewer #1: Yes

Reviewer #2: Yes

3. Have the authors made all data underlying the findings in their manuscript fully available?

Reviewer #1: No

Reviewer #2: Yes

 Raw data is now attached as a supplementary information excel file – S1 Raw data

4. Is the manuscript presented in an intelligible fashion and written in standard English?

Reviewer #1: Yes

Reviewer #2: Yes

5. Review Comments to the Author

Reviewer #1: Overview

The overall objective was very interesting, in comparing the three methods for detecting turtles that are already marked to maintain consistent monitoring effort of the population. I have suggested reorganising how you present the results, and clarifying that the surface observer result is an artefact of human bias, as this is not clear until the discussion, where it is beautifully presented. Many of the issues addressed are of broad interest to a general audience, however, the organisation of the Introduction limits this. I have suggested reorganizing how the information in this section is presented. I have advised adding more details to the methods to help clarify the different features of the approaches, while obvious to you, a general reader will not be able to perceive the different limits of each approach. I have suggested drawing on existing studies of sea turtles and other marine animals where some of the criteria delineated or described in methods and discussion have actually been addressed. Overall, this was a very interesting study, and a pleasure to read. Once these minor changes have been made, I believe this will make a very useful contribution to the journal.

Response: we thank the Reviewer for the positive and constructive comments, and have now reorganised it as suggested. 

Abstract

I would advise avoiding using the abbreviations for the three approaches, it is very hard to follow the context for general readers.

The authors feel that the UAV abbreviation is on common usage and not needing to be changed however both SO and UVW have been changed to Surface Observer and Underwater Video throughout except for UVW in figures where these abbreviations are now expanded in the caption.

Please clarify in an abstract what the marking is – this would impact detectability by the different methods.

Done 

Line 4

Nesters are first marked by painting their carapace with a longitudinal white stripe. Painted and unpainted turtles are then counted by a surface observer on a small boat in waters adjacent to the reef. 

Line 8 versus 16: I believe these results need to be reordered, 1, the detection rates and 2, the relative likelihood of resighting. Here, I would advise stressing that this is because from the surface a white marked turtle is more easily spotted than a non white marked turtle, making this approach biased to observer ability.

Done

Our results also demonstrated that surface observers consistently reported higher proportions of marked turtles than either the UAV or underwater video method. This in turn yielded higher population estimates with UAV or underwater video compared to the historical surface observer method, which suggested correction factors of 1.53 and 1.73 respectively. We attributed this to observer search error because a white marked turtle is easier to spot than the non-marked turtle.

Please check formatting specifications; I think PlosOne specify a single paragraph format.

Done

Introduction

PlosOne draws on a wide audience. I would advise starting the manuscript with the broad issue, i.e. the use of different methods to gauge population abundance – this is generic across marine and terrestrial species, with the issues of different technologies being ubiquitous.

Done

Population abundance is a fundamental metric underpinning wildlife management that is often quantified by the capture mark-recapture survey technique, which is based on the ratio of marked to unmarked animals in a population (Williams et al 2001). A less invasive and more cost-effective approach is the mark-resight approach where marked animals are subsequently visually identified without the need for physical recapture (McClintock and White 2009). Detection of animals with naturally identifying features or artificial marks can be enhanced by technologies such as unmanned aerial vehicles (UAVs) (Sorrel et al 2019), camera traps (Dunstan et al 2011), passive acoustics (Peel et al 2015) and telemetry (Lee et al 2014). For example, infrared camera traps have been used for mark-resight surveys of snow leopards using distinct pelage patterns (Jackson et al 2006). Yet the limitations are ubiquitous across marine and terrestrial species, as are the assumptions that must be met. Marked and unmarked animals must be similarly detectable irrespective of environmental conditions, marks must not influence the behaviour of the animal and marks must be stable over time (Williams et al 2001). It is therefore important to compare new methods against traditional approaches, not only in terms of the effectiveness of the approach but also in terms of whether assumptions are can be justified. This paper compares three mark-resight methods for estimating the abundance of nesting green turtles, Chelonia mydas: (1) a historically used surface-observer technique, (2) underwater video and unmanned aerial vehicle (UAV). 

Line 48, this is more methodological, In any case, clarify what is painted – numbers/codes or just colours? I would move this to the end of the introduction or start of methods.

Done

Line 132

The carapaces of nesting turtles were painted along the midline with a white stripe approximately 80 cm in length and 20 cm in width, using a 12 cm wide paint roller and “APCO-SDS fast dry water-based road marking paint” (MSDS Infosafe No. 1WDKY) (Dunstan, 2018)

Line 54, I would draw on this approach in a more general context and why it is better/more relevant than other approaches, identifying possible alternatives used on other wildlife. – i.e. I would make this my second paragraph, but more generic

Moved and changed but with addition rather than removing the more specific elements of LP estimation requirements – as detailed in next point below for text change

Line 64, good paragraph, this should be your second or third depending on how you plan to organise it, i.e. advances in data collection approaches followed by how to analyse it seems more logical…

Done

Line 64

The introduction of modern technologies such as UAVs and underwater video for counting surveys coupled with artificial intelligence for automated image analysis may provide a more time efficient and reliable mark-resight estimate. Another advantage of UAVs and underwater cameras compared to the vessel platform is that effect of surface reflections can be supressed or eliminated. 

The remoteness of Raine Island, the large in-water survey area required, the limitations of detectability throughout the turtle depth range, the sheer number of nesters on a given night and the high nesting failure and re-nesting by turtles over multiple nights have precluded total in-water or nesting censuses or a comprehensive mark recapture program. Instead, a mark-resight approach has been used to estimate the numbers of nesters in the surrounding internesting habitat since 1984 (Limpus et al 2003). Females are painted with a white longitudinal stripe on the carapace (marked) during nightly tally counts, and counts of marked and unmarked turtles in the waters that surround Raine Island are used to estimate abundance during the sampling period using the Lincoln-Petersen estimator (LP). Mark-resight is therefore combined with in-water sampling, and thus estimations of nester abundance are dependent on the limitations and assumptions of both approaches. 

Here we aimed to compare the effectiveness of the vessel observers, UAVs and underwater video, and to determine if the UAV and underwater camera estimates are comparable to the historical data. The major challenge for in-water surveys is to have high detectability for both marked and unmarked turtles, given that marine turtles spend only a small proportion of their time at the water surface, especially when surface conditions are poor, in turbid water or when turtles are amongst habitat structure (Fuentes et al 2015). The LP estimator is based on the assumption that the population is ‘closed’ during the sampling period (Williams et al 2002), which means that they do not depart from inter-nesting habitat in the short time interval from marking to the in-water survey. Our comparison of detectability of marked turtles between methods also provided the opportunity to test the key LP estimator assumption of equal detectability of marked and unmarked turtles. If the probability of detection is the same between marked and unmarked turtles, the ratio of marked and unmarked turtles should not differ between sampling methods. Finally, we used a repeated sampling study design to (a) determine whether there is a gain in precision in the LP estimator with repeated sampling, and (b) test whether the closure assumption was appropriate. The LP estimator is also only based on one resighting event, which could make it less robust than estimates from repeated sampling. 

Line 68, here start a new paragraph.

Done

Methods

Line 142, More details are required here. Please clarify the line of sight, number of observers and whether all 360 degrees was surveyed continuously. Please state whether transect lines were perpendicular or parallel to shore, the distance between transects and the total area covered.

Done

Line 173

A 4.2 m outboard powered rigid hull inflatable vessel with three persons aboard, one recording, one driving and one counting, was driven along the waters adjacent to the reef perimeter edge in search of the painted turtles. The survey track (Fig 1c) followed a general pattern around the reef perimeter with the vessel moving within 20m and parallel to the reef edge for approximately 100m then perpendicular to and away from the reef edge for 100m, parallel to and 120m from the reef edge for 100m and then perpendicular to and towards the reef edge for 100m. This pattern was repeated to complete a full circumnavigation survey of the reef. The single observer (Andrew Dunstan) was the same for all surveys. The observer was stationed at the bow of the vessel with a 180° line of sight and search area swathe of approximately 30m as the vessel moved forward at approximately 4 knots. This resulted in a total survey area of approximately 0.3km2.

Line 148, please state what the underwater field of view is, i.e. generally visibility is less compared to air, due to particles in the water.

Done

Line 185

Underwater video method (UWV). Underwater video surveys were conducted from the survey vessel simultaneously with the surface observer surveys (Table 1 and Fig1c & 2a). A GoPro Hero4 camera (frame rate: 25 hz; resolution: 1080; field of view: 127°) with an extended life battery was attached to the hull of the vessel pointing forward and downward, and recorded throughout the entire reef perimeter survey period. Underwater video visibility varied but was typically around 15m to provide a survey swathe of approximately 60m. Following the same track as the surface observer but with wider survey swathe resulted in a total survey area of approximately 0.4km2.

Line 164, why was this height selected? IN the figure, it shows that you flew it parallel to shore, how far offshore was the flight path? I would advise delineating this on the figure. Not just the path line, but the swathe.

Done

All points detailed in changed text

Line 204

UAV method. UAV surveys were conducted as close to midday as possible to reduce sun glare on the water surface. A DJI Inspire 1 UAV with Zenmuse X3 camera (frame rate: 25 hz; resolution: 1080; field of view: 94°) was flown at an altitude of 50 m and a speed of 5 m/s along a path parallel to the reef edge with the UAV vision swathe reaching from the reef edge to 90m oceanwards of the reef edge. (Figs 1c & 2b and S1 multimedia). This height was selected, after trials, to provide the ability to readily identify turtles in a variety of sea conditions while providing the broadest survey swathe possible. This camera and 20 mm equivalent lens provided a horizontal video survey swathe of 90 m at the sea surface. The UAV track resulted in a total survey area of approximately 0.6km2 Video footage was analysed as described for UWV surveys.

Phenomenal effort to collect data using all three approaches in tandem.

Thanks!

The white stripe makes the turtles extremely conspicuous; as this approach has been used for many years, is there a greater predation risk of these turtles compared to unmarked turtles? If you know the ratio of marked to unmarked turtles in the water and the ratio of marked to unmarked turtles with trauma/death, this would give some quantitative value. Also, with the drones, you might be able to view whether potential predators aggregate around marked turtles more.

There has been no predation of painted turtles observed and in fact very limited predation of live turtles observed. At Raine Island, sharks mostly scavenge turtles that have died on the intertidal areas and have subsequently floated off with the tide. 

Results

Line 205 – this should be just results, not results and discussion, as you present the discussion later.

Done

Line 206 - It would be useful/interesting to clarify for each approach, i.e. boat based observation – what was the greatest depth and distance of turtles observed by observers; underwater observation – same again, what was the greatest distance underwater and how deep; drones – same again, how deep could you detect turtles – were any on the seabed allowing you to determine this?

Our best attempt to do this was through using the turtle template and Secchi disc test. Further analysis could possibly determine depth and distance of detection using average size of a turtle combined with field of view calculations to provide distance from camera?? We will in the future look towards further clarification of these parameters for the different methods. The seabed at Raine reef surrounds (vertical reef edge) is from 50-300m deep and beyond detection by any of the methods.

This helps a reader understand the relative potential of each approach.

Line 277, good point, and nicely presented

Thanks

Line 279, agreed – also providing insights under conditions that would not be possible in a boat.

Thanks

Line 287, yes, but more detail is needed in the methods on the relative fields of view of the different methods to support this.

Done previously as above

Line 304, here, you should compare this result to that obtained by Schofield et al. It is interesting that the detection rates were not even higher for the drone:

Schofield G, Katselidis KA, Lilley MKS, Reina R, Hays G. 2017. Detecting elusive aspects of wildlife ecology using UAVs: new insights on the mating dynamics and operational sex ratios of sea turtles. Functional Ecology 31 (12), 2310–2319 DOI: 10.1111/1365-2435.12930

Hodgson, A., Kelly, N., Peel, D. 2013. Unmanned aerial vehicles (UAVs) for surveying marine fauna: a dugong case study. PLOS ONE 8: e7955

Yes, line 304 relates to the population estimate differences rather than total turtles sighted/counted by each method. The UAV does sight more turtles per survey area than the other techniques – Line 291. 

Line 305, this implies that the underwater approach obtained more turtles than the drone? Is there any explanation of why?

This was an error that has now been corrected. Thanks bringing this to our attention. UAVs yielded a population estimate 1.73x higher than the historical surface observer method, whereas the underwater video method estimated 1.53x more turtles than the surface observer method.

 Is the underwater visibility exceedingly good at your study location. I know that at other locations, for it to exceed 5 m is rare. This should be stressed that this result might differ to other locations where underwater visibility is not so clear.

Done

to be driven by the extreme variability in the density of turtles in the inter-nesting habitat, suggesting that robust correction factors would require more sampling across a range of turtle densities. The oceanic waters surrounding Raine Island are extremely clear with visibility greater than 10m and often more than 20m. This may impose limitations for adapting this methodology to locations with lower visibility especially inshore habitats.

Line 311, yes, absolutely. The observer would also start only looking for white strips due to it being easier. This needs to be clarified in the Abstract too, because, at present the way it is presented implies a robust/objective output, rather than observer issues.

Added to Abstract

Line 15

This in turn yielded higher population estimates with UAV or underwater video compared to the historical surface observer method, which suggested correction factors of 1.53 and 1.73 respectively. We attributed this to observer search error because a white marked turtle is easier to spot than the non-marked turtle. In contrast, the UAV and underwater video methods allowed subsequent frame-by-frame review, thus reducing observer search error.

Line 316, absolutely.

Thanks

Line 321, yes, very good observation/reference.

Thanks

Line 331, it would be good to present this in methods/results for all three approaches.

Inserted to start of results section and presented previously within methods section

Line 252

The white mark was discernible at an average of 3 metres deeper than the unpainted turtle model (t = 3.61, df = 3.8, p = 0.026). The in-water detectability of painted and unpainted turtles indicated that turtles were identifiable to 10 m depth, and that there were no pronounced differences in water clarity between sampling locations that were likely have influenced the results.

Line 351, this should be your first key finding presented in the abstract.

Inserted to abstract

Line 8

Here we compare and evaluate the three methods. We found comparatively little variation in resighting probabilities between consecutive days of sampling or time of day, which supports an underlying assumption of the method (i.e. demographic closure during sampling). This lack of bias in the location availability for detection of painted versus unpainted turtles and further supported by a parallel satellite tracking study of 40 turtles at Raine Island.

Line 351, paragraph – there are several studies on monitoring sea turtles with drones that have been published and should be referred to/compared here. See the following paper for a review of the various papers on which to draw information:

Schofield G, Esteban N, Katselidis KA, Hays GC. 2019. Drones for research on sea turtles and other marine vertebrates – A review. Biological Conservation https://doi.org/10.1016/j.biocon.2019.108214

Bevan, E., Wibbels, T., Najera, B.M., Martinez, M.A. et al., 2015. Unmanned aerial vehicles (UAVs) for monitoring sea turtles in near-shore waters. Mar. Turtle Newsl. 145, 19−22

Inserted reference text to Discussion

Line 432

Although no other studies have used UAVs in conjunction with mark-resight to estimate turtle abundance (Schofield et al 2019),

Other turtle abundance studies are further referenced to augment the study of (Sykora-Bodie et al 2017).

Line 432

Although no other studies have used UAVs in conjunction with mark-resight to estimate turtle abundance (Schofield et al 2019), other UAV studies have used the direct count method for in-water (Schofield et al., 2017; Sykora-Bodie et al., 2017; Hensel et al., 2018) and operational sex ratios (Schofield et al., 2017). Aerial surveys for nesting beach track counts (Witt et al 2009) may be more effectively undertaken by UAVs in the future. In-water UAV abundance counts of turtles are adjusted for the availability bias (Schofield et al 2019), however these adjustments were not deemed necessary in the Raine Island study due to the clear waters allowing detection to at least a 10 m depth range.

Line 364 – these aspects have been justified by previous studies, so it is worth drawing on these to support these decisions.

Reference to a comprehensive techniques paper

Line 423

Methods for manually counting wildlife from video footage are well documented (Hollings et al 2018). The use of video recording versus the use of overlapping still images to produce a single orthomosaic image by UAV were both considered. For this application the

Line 372, again, see the following paper where the potential detectability of turtles at different seabed depths was investigated

Schofield G, Katselidis KA, Lilley MKS, Reina R, Hays G. 2017. Detecting elusive aspects of wildlife ecology using UAVs: new insights on the mating dynamics and operational sex ratios of sea turtles. Functional Ecology 31 (12), 2310–2319 DOI: 10.1111/1365-2435.12930

Yes, changed in discussion as below

Line 391

The in-water detectability of painted and unpainted turtles indicated that turtles were identifiable to 10 m depth from UAV height of 50m, and that there were no pronounced differences in water clarity between sampling locations that were likely have influenced the results. These results are similar to those previously reported (Schofield et al 2017) where UAV flights at 60m were the maximum height for reliable detection of turtles in an inshore habitat to a depth of 7m at the seabed.

Figure 2 – I would advise adding a photo of boat based observation too to present all three approaches here.

Good point – done and caption adjusted

Figure 2. Underwater video and UAV survey image examples. (a) Still image from drone survey in conjunction with surface observer and underwater video surveys showing survey vessel and (b) Still image from underwater video December 2017 survey and (c) still image from UAV video survey December 2017 at 50 m survey altitude.

Reviewer #2: A few minor comments to improve this nice, concise manuscript.

- In the introduction, I suggest that some more context for use of UAVs is provided, e.g. the sorts of uses outlined in Rees et al 2018 The potential of unmanned aerial systems for sea turtle research and conservation: a review and future directions. Endangered Species Research 35: 81-100.

Good point. Inserted into Introduction

Line 69

A recent review of the uses of UAVs in marine and turtle research and management (Rees et al 2018) identifies the broad scope of opportunities and benefits possible. The capacity to increase efficiency, reduce field personnel exposure to risks and provide new and/or better quality data gathering options, not just for population estimation, is detailed. UAVs can benefit studies on turtle nesting (including night monitoring with thermal cameras), at-sea and foraging area distribution surveys, a wide range of behavioural studies, surveillance against illegal take and hatchling dispersal and survivorship. UAVs also provide more efficient and higher resolution methods for mapping and topographic profiling of key turtle nesting and foraging habitats. 

- analysis of UAV and UWV video: was there any analysis done of learning effect to account for the viewer becoming more expert at detecting turtles as more video was observed?

No, but we are working towards AI and will be doing this during ground-truthing and testing the AI results

- was a polarising or UV filter used on the UAV camera?

Yes, a polarising filter – added in methods

Line 204

UAV method. UAV surveys were conducted as close to midday as possible to reduce sun glare on the water surface. A DJI Inspire 1 UAV with Zenmuse X3 camera (frame rate: 25 hz; resolution: 1080; field of view: 94°; polarising filter)

- delete 'on' line 338

Done

- author name is missing in reference 23

Yes, was doubled up. Now removed

- general comment for the discussion: is there opportunity for automated video analysis by image search of UAV and UWV video by computer?

Yes, already in progress and in final trail stages. Working well.

- notwithstanding the assumption that they are absent, were males ever observed by any of the survey techniques?

No, but they may have been overlooked

6. PLOS authors have the option to publish the peer review history of their article (what does this mean?). If published, this will include your full peer review and any attached files.

Do you want your identity to be public for this peer review? For information about this choice, including consent withdrawal, please see our Privacy Policy.

Reviewer #1: No

Reviewer #2: Yes: Richard Reina

---

## [Decision Letter · Decision Letter 1]

7 May 2020

Use of unmanned aerial vehicles (UAVs) for mark-resight nesting population estimation of adult female green sea turtles at Raine Island

PONE-D-20-01406R1

Dear Dr. Dunstan,

We are pleased to inform you that your manuscript has been judged scientifically suitable for publication and will be formally accepted for publication once it complies with all outstanding technical requirements.

With kind regards,

Graeme Hays

Academic Editor

PLOS ONE

Additional Editor Comments (optional):

The authors have made a good effort to revise the manuscript in line with the comments. Thank you to the authors for attending to the comment so thoroughly. I think this manuscript can now be accepted for publication in PLoS1. It will make a nice contribution. Graeme Hays

Reviewers' comments:

Reviewer's Responses to Questions

**Comments to the Author**

1. If the authors have adequately addressed your comments raised in a previous round of review and you feel that this manuscript is now acceptable for publication, you may indicate that here to bypass the “Comments to the Author” section, enter your conflict of interest statement in the “Confidential to Editor” section, and submit your "Accept" recommendation.

Reviewer #1: All comments have been addressed

Reviewer #2: All comments have been addressed

2. Is the manuscript technically sound, and do the data support the conclusions?

Reviewer #1: Yes

Reviewer #2: Yes

3. Has the statistical analysis been performed appropriately and rigorously? 

Reviewer #1: Yes

Reviewer #2: Yes

4. Have the authors made all data underlying the findings in their manuscript fully available?

Reviewer #1: Yes

Reviewer #2: Yes

5. Is the manuscript presented in an intelligible fashion and written in standard English?

Reviewer #1: Yes

Reviewer #2: Yes

6. Review Comments to the Author

Reviewer #1: The authors have addressed all comments sufficiently.

I look forward t seeing the final publication.

Reviewer #2: (No Response)

7. PLOS authors have the option to publish the peer review history of their article (what does this mean?). If published, this will include your full peer review and any attached files.

Reviewer #1: No

Reviewer #2: No

---

## [Editor Report · Acceptance letter]

15 May 2020

PONE-D-20-01406R1 

Use of unmanned aerial vehicles (UAVs) for mark-resight nesting population estimation of adult female green sea turtles at Raine Island 

Dear Dr. Dunstan:

I am pleased to inform you that your manuscript has been deemed suitable for publication in PLOS ONE. Congratulations! Your manuscript is now with our production department. 

With kind regards,

on behalf of

Professor Graeme Hays 

Academic Editor

PLOS ONE